# Cross-Reactivity of Palladium in a Murine Model of Metal-induced Allergic Contact Dermatitis

**DOI:** 10.3390/ijms21114061

**Published:** 2020-06-05

**Authors:** Hiroaki Shigematsu, Kenichi Kumagai, Motoaki Suzuki, Takanori Eguchi, Ryota Matsubara, Yasunari Nakasone, Keisuke Nasu, Takamasa Yoshizawa, Haruno Ichikawa, Takahiro Mori, Yoshiki Hamada, Ryuji Suzuki

**Affiliations:** 1Department of Oral and Maxillofacial Surgery, Toshiba Rinkan Hospital, 7-9-1 Kamitsuruma, Minami-ku, Sagamihara, Kanagawa 252-0385, Japan; shigematsu_hiroaki@yahoo.co.jp; 2Department of Oral and Maxillofacial Surgery, School of Dental Medicine, Tsurumi University, 2-1-3 Tsurumi, Tsurumi-ku, Yokohama, Kanagawa 230-8501, Japan; eguchi-t@tsurumi-u.ac.jp (T.E.); pd19008@stu.tsurumi-u.ac.jp (K.N.); pd19010@stu.tsurumi-u.ac.jp (T.Y.); ichikawa-h@tsurumi-u.ac.jp (H.I.); hamada-y@tsurumi-u.ac.jp (Y.H.); 3Department of Clinical Immunology, Clinical Research Center for Rheumatology and Allergy, Sagamihara National Hospital, National Hospital Organization, 18-1 Sakuradai, Minami-ku, Sagamihara, Kanagawa 252-0392, Japan; mtakszk@gmail.com (M.S.); suzuki.ryuji.aj@mail.hosp.go.jp (R.S.); 4Graduate School of Life and Environmental Sciences, University of Tsukuba, 1-1-1, Tennodai, Tsukuba, Ibaraki 305-8577, Japan; 5Department of Oral and Maxillofacial Surgery, Saku General Hospital, 197, Usuda, Saku, Nagano 394-0301, Japan; ryota_matsubara_gtonty@yahoo.co.jp; 6Department of Oral and Maxillofacial Surgery, Nagano Matsushiro General Hospital, 183, Matsushiro, Nagano 381-1231 Japan; ynakasone@hosp.nagano-matsushiro.or.jp; 7Departments of Clinical Oncology and Gastroenterological Surgery, National Hospital Organization Sagamihara National Hospital, National Hospital Organization, 18-1 Sakuradai, Minami-ku, Sagamihara, Kanagawa 252-0392, Japan; westendc7m@gmail.com

**Keywords:** metal allergy, contact dermatitis, metal-specific T cells, cross-reactivity, T-cell receptor, palladium, nickel, chromium

## Abstract

Metal allergy is usually diagnosed by patch testing, however, the results do not necessarily reflect the clinical symptoms because of cross-reactivity between different metals. In this study, we established the novel mouse model of cross-reactive metal allergy, and aimed to elucidate the immune response in terms of T-cell receptor repertoire. This model was classified into two groups: the sensitization to nickel and challenge with palladium group, and the sensitization to chromium and challenge with palladium group. This model developed spongiotic edema with intra- and peri-epithelial infiltration of CD4^+^ T cells in the inflamed skin that resembles human contact dermatitis. Using T cell receptor analysis, we detected a high proportion of T cells bearing *Trav8d-1-Traj49* and *Trav5-1-Traj37* in the Ni- and Cr-sensitized Pd-challenged mice. Furthermore, mucosal-associated invariant T cells and invariant natural killer T cells were also detected. Our results indicated that T cells bearing *Trav8d-1-Traj49* and *Trav5-1-Traj37* induced the development of palladium-cross reactive allergy, and that mucosal-associated invariant T and invariant natural killer T cells were also involved in the cross-reactivity between different metals.

## 1. Introduction

Metal allergy is defined as a delayed-type hypersensitivity reaction triggered by the repeated contact of an antigen to the skin [1]. This reaction is mediated by antigen-specific T cells accumulating in the inflamed skin [1]. The number of patients with a metal allergy has increased recently because of the greater chance for metal exposure related to the increasing use of metal accessories such as piercings, and the advancement of medical technology, such as prosthetic and implant treatment in dentistry and joint replacement in orthopedic surgery [2,3,4,5]. In addition to nickel (Ni), palladium (Pd) and chromium (Cr) were reported as causal metals for allergic contact dermatitis [6,7]. Dental materials often contain Pd because of its resistance to corrosion; therefore, metal allergy caused by Pd eluted from dental materials has become a serious problem [3].

We previously developed novel murine models of Pd, Ni, and Cr allergy by sensitization with Pd, Ni, or Cr plus lipopolysaccharide (LPS) solution into the groin, and following challenges by injecting these metal solutions into the footpads [8,9,10]. These models represent delayed-type hypersensitivity responses of metal allergy and allow the investigation of infiltrating T cells in the elicitation phase. We analyzed the T cell receptor (TCR) repertoire of infiltrating T cells in the inflamed skin, and found metal-specific T cells in Pd and Cr allergy. Interestingly, innate lymphoid cells (ILCs) such as invariant natural killer T (iNKT) cells were also detected in the Ni, Cr, and Ti allergic mouse models [8,9,10].

In the clinical setting, metal allergy is mainly diagnosed by patch testing; however, the results do not necessarily reflect the clinical symptoms because of the cross-reactivity between different metals [11]. Most previous studies have investigated T cell clones isolated from patients with cross-reactive metal allergy [12,13]. Another study investigating whether Ni-specific T cells isolated from patients with Ni allergy cross-reacted with transition metals (e.g., copper or Pd) suggested that co-reactivity in vivo might be a consequence of the cross-reactive nature of some T cell clones [14]. However, because of the unavailability of suitable cross-reactive metal allergy animal models, how pathogenic T cell clones at the sites of allergic inflammation contribute to the development of cross-reactivity in metal allergy has not been explored.

In the present study, to elucidate the immune response of cross-reactive metal allergy, we have established the novel mouse model of cross-reactive metal allergy, and characterized footpad-infiltrating T cells during the elicitation phase in terms of phenotypic T cell markers, TCR repertoires, and cytokine expressions.

## 2. Results

### 2.1. Footpad Swelling in a Metal Allergy Cross-reaction Mouse Model

All experimental protocols were undergone as described in the Materials and Methods. To determine how accumulated inflammatory cells in inflamed skin contribute to the development of cross-reactive metal allergy, we generated a novel mouse model of cross-reactive metal allergy. Footpad swelling was measured at 1, 3, and 7 days after the third challenge with a metal solution injection into the footpad. The size of the swelling peaked 1 day after the third challenge in all groups (Figure 1). The footpad swelling was more intense in the Ni-Pd group than in the Cr-Pd group at 1 day after the final challenge. We used saline-sensitized and Pd-challenged mice (Saline-Pd group) as controls. At 7 days after the third challenge, the footpad swelling in the Ni-Pd and Cr-Pd groups was significantly increased compared with Saline-Pd group. Footpad swelling was not significantly different between Ni-Pd and Cr-Pd groups. As shown in a representative image, swelling and crust formation of the footpad were observed at 7 days after the third challenge in the Ni-Pd and Cr-Pd groups (Figure 2).

### 2.2. Histological and Immunohistochemical (IHC) Analyses of T Cell Markers in the Footpads of Metal Allergy Cross-Reaction Mice

To verify whether T cells infiltrated into inflamed skin, we analyzed the footpad skin of metal allergy cross-reaction mice and saline-treated mice at 7 days after the third challenge. Hematoxylin and Eosin (H&E) staining showed epithelial acanthosis, as well as epidermal spongiosis and liquefaction degeneration in the Ni-Pd and Cr-Pd groups (Figure 3). Immunohistochemical (IHC) staining showed that CD3^+^ T cells were also present in the epithelial basal layer and the upper dermis in the Ni-Pd and Cr-Pd groups (Figure 3). In contrast, inflammatory reactions were not observed in the footpads of the Saline-Pd group (Figure 3). IHC analysis of CD4 and CD8 expressions on CD3^+^ T cells that had infiltrated into inflamed skin in the Ni-Pd and Cr-Pd groups showed that CD4^+^CD8^−^ T cells were present in the epithelial basal layer and the upper dermis (Figure 4).

### 2.3. Expression of Inflammatory Cell Markers in the Metal Allergy Cross-reaction Mouse Model

We compared the expression levels of Th1 cytokines (interferon gamma [IFN-γ], tumor necrosis factor alpha [TNF-α], and interleukin [IL]-2), Th2 cytokines (IL-4, IL-5, and IL-6), cytotoxic granules (granzyme A and B), and apoptosis-related genes (Fas and Fas ligand [L]) by qPCR analysis in inflamed footpads of the Ni-Pd, Cr-Pd, and Saline-Pd groups at 7 days after the third challenge. The expression levels of TNF-α, IL-4, IL-5, and cytotoxic granules (granzyme A and B) were significantly higher in the Ni-Pd and Cr-Pd groups than in the Saline-Pd group (Figure 5). The expression levels of inflammatory cell markers were not significantly different between Ni-Pd and Cr-Pd groups.

### 2.4. T Cell Receptor (TCR) Repertoire Usage in the Metal Allergy Cross-reaction Mouse Model

To examine the TCR repertoire of T cells infiltrating the footpads of metal allergy cross-reaction mice, we analyzed the TRAV and TRAJ expression levels and CDR3 sequence in inflamed footpads from the Ni-Pd and Cr-Pd groups at 7 days after the third challenge using next-generation sequencing (NGS) [15]. TRAV and TRAJ expression levels and CDR3 sequence analyses showed that a high proportion of T cells expressed *Trav8d-1-Traj49* and *Trav5-1-Traj37* in the Ni-Pd and Cr-Pd groups (Figure 6). In addition, some shared T cells bearing the same TRAV, TRAJ, and CDR3 sequences were obtained from the Ni-Pd and Cr-Pd groups (Figure 6). Interestingly, mucosal-associated invariant T (MAIT) cells and iNKT cells were observed in both groups (Figure 6). It was reported that MAIT cells express Trav1-Traj33 while iNKT cells express Trav11d-Traj18 [16,17].

## 3. Discussion

In the current study, we established a novel murine model of Pd-induced cross-reactive metal allergy in which T cells infiltrated the inflamed area. The model developed spongiotic edema with intra- and peri-epithelial infiltration of CD4^+^ T cells in the inflamed skin that resembles human contact dermatitis. Infiltrating T cells in the Ni- and Cr-sensitized Pd-challenged mice expressed CD4^+^ and secreted cytotoxic granules. Of note, they used a common TCR repertoire expressing *Trav8d-1-Traj49* and *Trav5-1-Traj37*. Furthermore, MAIT and iNKT cells were also detected. This study is the first report to examine the immune response of cross-reactive metal allergy in inflamed skin mediated by specific TCR-expressing T cells and innate lymphoid cells.

Pd and Cr induce contact dermatitis in patients with Ni allergy [18]. Pd is in the same group as Ni, and Cr is in the same period as Ni in the periodic system. In particular, the simultaneous positive reactions of Ni and Pd are explained because Ni and Pd form similar chemistry and electron arrangements [11,19]. In addition, Santucci et al. reported that Ni and Pd form similar complexes with sulfur ligands, which might explain why both metals form comparable metal-protein complexes [20]. Because the mechanism involved in the cross-reactivity between Cr and Pd is still unclear, further research is needed.

A previous study suggested that metal allergy is associated with either CD4^+^ or CD8^+^ T cell activation depending on the antigen processing pathway involved [21,22]. Hirai et al. demonstrated that the administration of metallic nanoparticles induced ear swelling that was reduced by eliminating CD4^+^ T cells [23]. These data indicated that metal allergy is a CD4^+^ T cell-dependent immune response. In the current study, a large number of CD4^+^ T cells were detected in the metal allergy cross-reactive mouse model.

In support of our findings regarding the role of T cells, Pd-challenged mice had significantly higher expression levels of TNF-α, IL-4, IL-5, and granzyme, which is a functional molecule produced by cytotoxic T cells [24]. Kitagaki et al. reported that the repeated elicitation of contact hypersensitivity induced a shift in cutaneous cytokine milieu from a T helper cell type 1 to a T helper cell type 2 profile. [25]. Our results were consistent with previous studies, thus the repeated challenges may affect Th2 immune response as well as Th1. A previous study reported that epidermal cell apoptosis was induced by infiltrating T cells that released Fas-L [26]. We previously reported the involvement of Fas and Fas-L in mouse Pd and Cr allergy models, and increased granzyme, Fas, and Fas-L in mouse Ni allergy models [8,9,10]. Therefore, the results of the current study are consistent with our previous reports. The observed data indicated that the inductive mechanism of apoptosis might vary according to the metal used in the model.

We used NGS to analyze TRAV and TRAJ expression levels and the common CDR3 sequences of TRA. NGS allows a comprehensive quantitative analysis of TCR repertoires at the clonal level and is also suitable for the identification of antigen-specific T cells in the metal allergy cross-reaction mouse model. We previously reported that metal-specific immune responses were observed in murine models of Ni-, Pd-, Cr-, and Ti-induced allergic contact dermatitis [8,9,10,27]. This suggests that accumulated metal-specific T cells differ according to the type of metal applied at the site of allergic inflammation. In Pd-challenged mice, we detected a high proportion of T cells expressing *Trav8d-1-Traj49* and *Trav5-1-Traj37*. Furthermore, high numbers of MAIT and iNKT cells were also identified. MAIT and iNKT cells express an invariant TCR chain that was discovered during the analysis of human CD4^−^CD8^−^ T cells [28]. Subsequently, Lantz et al. observed the abundant presence of Vα19i T cells in the lamina propria and Peyer’s patches of the intestines in mouse homolog experiments [29,30]. These cells were associated with immunologic diseases. While low levels of MAIT cells are present in the peripheral blood, they accumulate in inflammatory tissues in many immune disorders. Chiba et al. reported that MAIT cells promoted arthritis progression in an inflammatory arthritis mouse model [31]. Kumagai et al. showed that iNKT cells were an important regulator of metal allergy [32], and that MAIT cells were involved in Ti allergy murine models [27]. iNKT and MAIT cells were detected in the present study, suggesting they might have a role in the development of cross-reactive metal allergy.

In contrast to most other haptens that covalently bind to proteins, metals interact with electron-rich atoms via coordinate bonds. The number of ligands surrounding a metal ion, the “coordination number”, leads to a preferential spatial arrangement of these ligands [33]. Each metal complex, according to the oxidation status of the metal, is usually characterized by the nature of these ligands as well as its preferred coordination geometry. Moulin et al. reported that the cross-reactivity between Pd and Ni might be explained by different binding affinities of metals to specific binding motifs of proteins including MHC molecules and MHC-associated peptides [34]. Our TCR repertoire data contribute to the further understanding of the structural identity of antigenic determinants recognized by cross-reactivity between Pd, Ni, and Cr-induced T cells. Moreover, our results can provide a platform for the development of new diagnostic, and treatment approaches relevant to the resolution of metal-induced cross reactivity. The collective genetic information of cross-reactive T-cells will guide further investigations of metal allergy, and strengthen the safety standards in dental and medical treatments.

In conclusion, we demonstrated that CD4^+^ cells infiltrated into the site of cross-reactive allergic inflammation in a murine metal allergy model, and established Pd-cross-reactive specific T-cell clones. These results suggest that restricted T cells bearing *Trav8d-1-Traj49* and *Trav5-1-Traj37* have the potential for specific recognition in Pd cross-reactive allergy. The direct cloning of TCR genes from local sites of inflammation using this model would be a useful tool for understanding T cell mediated immune disease in cross-reactive metal allergy, as well as providing new insights into cross-reactive metal allergy.

## 4. Materials and Methods

### 4.1. Ethics Statement

This study was performed in a strict accordance with recommendations in the Guidelines for Care and Use of Laboratory Animals set by the Tsurumi University Japan (approval number: 28P046, issued on 13 April 2017). All animal experiments were performed according to the relevant ethical requirements and with approval from the committees for animal experiments at Tsurumi University, Japan. All surgeries were performed under three types of mixed anesthetic agents and all efforts were made to minimize the suffering of animals. To ensure death and prevent pain caused by tissue harvesting, all mice were sacrificed by cervical dislocation under three types of mixed anesthetic agents.

### 4.2. Animals

BALB/cAJcl mice (5-week-old females, *n* = 15) were obtained from CLEA Japan (Tokyo, Japan). Mice were maintained in standard aluminum cages (with a lid made of stainless-steel wire). Food and water were available ad libitum.

### 4.3. Reagents

Palladium(II) chloride (PdCl_2_), chromium(II) chloride (CrCl_2_), and nickel(II) chloride (NiCl_2_) were purchased from Wako Pure Chemical Industries (Osaka, Japan). Lipopolysaccharide (LPS) from Escherichia coli (O55:B5), prepared by phenol–water extraction, was purchased from Sigma (St. Louis, MO, USA). PdCl_2_, CrCl_2_, NiCl_2_, and LPS were dissolved in sterile saline.

### 4.4. Anesthetic Agents

The anesthetic was prepared as a mixture of three drugs. Medetomidine hydrochloride was purchased from Nippon Zenyaku Kogyo Co., Ltd. (Fukushima, Japan), midazolam was purchased from Sandoz (Tokyo, Japan) and butorphanol tartrate was purchased from Meiji Seika Pharma Co., Ltd. (Tokyo, Japan). These drugs were kept at room temperature (RT). We mixed medetomidine hydrochloride at a dose of 0.3 mg/kg, midazolam at a dose of 4 mg/kg, and butorphanol tartrate at a dose of 5 mg/kg. The concentration ratio of the three types of mixed anesthetic agents was determined according to a previous study [35]. Usually, 0.75 mL of medetomidine hydrochloride, 2 mL of midazolam, 2.50 mL of butorphanol tartrate, and 19.75 mL of sterile saline were mixed to make 25 mL of anesthetic agent. All agents were diluted in sterile saline and stored at 4 °C in the dark. The mixed anesthetic agents were administered to mice at a volume of 0.01 mL/g of body weight. All mice were injected intraperitoneally with the mixture of the three types of anesthetic agents.

### 4.5. Development of the Metal Allergy Cross-reaction Mouse Model

We established the experimental protocols for the metal allergy cross-reaction mouse model based on previous reports [8,9,10] (Figure 7). All experiments were carried out in another room after transfer from the animal holding room.

Sensitization: In ten BALB/c mice, 125 µL of a 10 mM metal solution (CrCl_2_ or NiCl_2_) with 10 µg/mL LPS were injected twice (at an interval of 7 days) intradermally (i.d.) into the left and right groin of mice (250 µL each). At 7 days after sensitization, mice were challenged for the first time.

Challenge for elicitation: After two rounds of sensitization, a challenge to elicit an immune response was performed using a 10 mM metal solution (PdCl_2_) with 25 µL injected i.d. each into the left and right footpads under anesthesia with three types of mixed anesthetic agents. The challenge was repeated three times at an interval of 14 days.

Metal allergy cross-reaction mice were classified into two groups: sensitization to NiCl_2_ with LPS and challenge with PdCl_2_ (Ni-Pd) group (*n* = 5) and sensitization to CrCl_2_ with LPS and challenge with PdCl_2_ (Cr-Pd) group (*n* = 5). Mice which were sensitized with saline and challenged with PdCl_2_ were used as the control sample (Saline-Pd group, *n* = 5).

### 4.6. Measurement of Footpad Swelling

Swelling of the plantar region was measured at 1, 3, and 7 days after the third injection using a Peacock Dial Thickness Gauge (Ozaki MFG Co. Ltd., Tokyo, Japan).

### 4.7. Immunohistochemical Analyses

Footpads were obtained from metal allergy cross-reaction mice for histology and IHC analyses. Tissue samples were fixed with 4% paraformaldehyde-lysine-periodate overnight at 4 °C. After washing with phosphate buffered saline (PBS), fixed tissues were penetrated by soaking in 5% sucrose/PBS for 1 h, 15% sucrose in PBS for 3 h, and then 30% sucrose in PBS overnight at 4 °C. Tissue samples were embedded in Tissue Mount (Chiba Medical, Saitama, Japan) and snap-frozen in a mixture of acetone and dry ice. Frozen sections were sliced into 6-µm cryosections and air-dried on poly-L-lysine-coated glass slides. For histological analyses, the cryosections were stained with H&E. For IHC analyses, antigen retrieval was performed. The cryosections were then stained with anti-mouse CD3 (SP7, Abcam), CD4 (H129.19, Pharmingen, San Diego, CA, USA), and CD8α (53-6.7, Pharmingen) monoclonal antibodies (mAbs). Non-specific binding of mAbs was blocked by incubation of sections with PBS containing a 1:20 dilution of normal goat serum or normal rabbit serum, 0.025% Triton X-100 (Wako Pure Chemicals) and 5% BSA (Sigma–Aldrich) for 30 min at room temperature (RT). The sections were incubated with primary mAbs for 1 h at RT. After washing three times with PBS for 5 min, intrinsic peroxidase was quenched by 3% H_2_O_2_ in methanol. After soaking the sections in distilled water, they were washed twice. Sections were then incubated with a secondary antibody (biotinylated goat anti-hamster IgG antibody or biotinylated rabbit anti-rat IgG antibody) for 1 h at RT. After washing three times, the sections were incubated with Vectastain ABC Reagent (Vector Laboratories, Burlingame, CA, USA) for 30 min at RT, followed by 3,3′-diaminobenzidine (DAB) staining (0.06% DAB and 0.03% H_2_O_2_ in 0.1 M Tris-HCl buffer pH 7.6, Wako Pure Chemicals). Finally, the tissue sections were stained with hematoxylin to visualize the cell nuclei.

### 4.8. Isolation of Total RNA from Tissues

Fresh footpads were obtained from mice and immediately soaked in RNAlater RNA Stabilization Reagent (Qiagen, Hilden, Germany). Total RNAs from footpads were extracted using the RNeasy Lipid Tissue Mini Kit (Qiagen) according to the manufacturer’s instructions.

### 4.9. Expression of Inflammatory Cell Markers

The expression levels of immune response-related genes including Th1 cytokines (IFN-γ, TNF-α, and IL-2), Th2 cytokines (IL-4, IL-5, and IL-6), cytotoxic granules (granzyme A and B), and apoptosis-related genes (Fas and Fas ligand) were measured by quantitative polymerase chain reaction (qPCR) using the Bio-Rad CFX96 system (Bio-Rad, Hercules, CA, USA). Freshly isolated total RNA from the footpads of mice was converted to cDNA using PrimeScript™ RT Reagent Kit (Takara Bio) according to the manufacturer’s instructions. The PCR reaction consisted of 5 µL of SsoFast™ EvaGreen Supermix (Bio-Rad), 3.5 µL of RNase/DNase-free water, 0.5 µL of 5 µM primer mix, and 1 µL of cDNA in a final volume of 10 µL. Cycling conditions were as follows: 30 s at 95 °C followed by 45 rounds of 1 s at 95 °C and 5 s at 60 °C. At the end of each run, melting curve analyses were performed from 65 °C to 95 °C to confirm the homogeneity of PCR products. All assays were repeated three times and the mean values were used for gene expression levels. Five points of tenfold serial dilutions of each standard transcript were used to determine the absolute quantification, specification, and amplification efficiency of each primer set. Standard transcripts were generated by in vitro transcription of the corresponding PCR product into a plasmid. The nucleotide sequences were confirmed by DNA sequencing using the CEQ8000 Genetic Analysis System (Beckman Coulter, Fullerton, CA, USA). Sequence quality and concentration were validated using the Agilent DNA 7500 Kit in an Agilent 2100 Bioanalyzer (Agilent, Santa Clara, CA, USA). *GAPDH* gene expression was used as an internal control. The expression levels of each target gene were normalized to *GAPDH* expression.

### 4.10. TCR Data Analyses

Total RNA was prepared from the footpads of the Ni-Pd and Cr-Pd group mice at day 7 after the third challenge and converted to cDNA with Superscript III reverse transcriptase (Invitrogen, Carlsbad, CA). Next, unbiased adaptor-ligation PCR was used to amplify the TCR genes [15]. High-throughput sequencing was performed with the Illumina MiSeq paired-end platform (2 × 300 bp) (Illumina, San Diego, CA, USA). TRAV and TRAJ segments in TCR genes were assigned a data set of reference sequences from the international ImMunoGeneTics information system (IMGT) database (http://www.imgt.org). Data processing, assignment, and data aggregation were automatically performed using repertoire analysis software originally developed by our group (Repertoire Genesis, Osaka, Japan). The amino acid sequences of CDR3 regions ranged from a conserved cysteine at position 104 of the IMGT nomenclature to a conserved phenylalanine at position 118 with the following glycine translated from the nucleotide sequences. A unique sequence read was defined as a sequence read having no identity in TRAV and TRAJ and a deduced amino acid sequence of CDR3 with other sequence reads. The copy number of identical unique sequence reads in each sample was automatically counted by repertoire analysis software and then ranked in order of the copy number. Percentage occurrence frequencies of sequence reads with TRAV and TRAJ, and genes in total sequence reads were calculated.

### 4.11. Statistical Analysis

Statistically significant differences between the mean values of each experimental group were analyzed using the Kruskal–Wallis test followed by Dunn’s multiple comparison tests using GraphPad Prism 7 software for Windows (GraphPad Software Inc., San Diego, CA, USA). A *p* value < 0.05 was statistically significant.

## Figures and Tables

**Figure 1 ijms-21-04061-f001:**
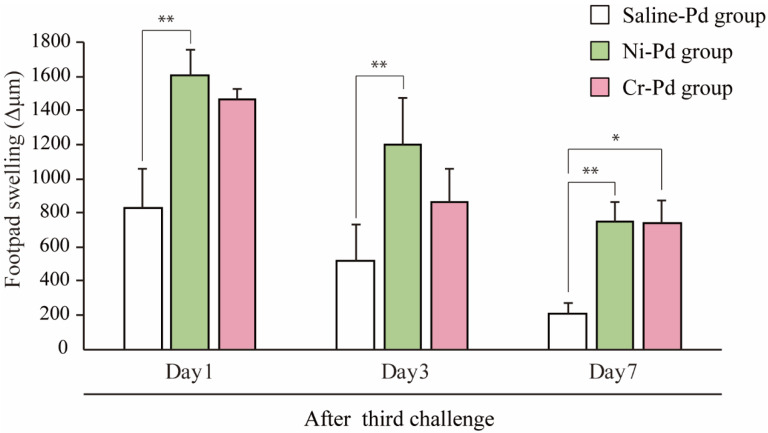
Footpad swelling in the mouse model of cross-reactive metal allergy. Footpad swelling was measured at various time points. The Cr-Pd, Ni-Pd, and Saline-Pd groups were analyzed at 1, 3, and 7 days after the third challenge. Bars and error bars indicate the mean ± SD. Statistical significance was tested by the Dunn’s test (* *p* < 0.05, ** *p* < 0.01).

**Figure 2 ijms-21-04061-f002:**
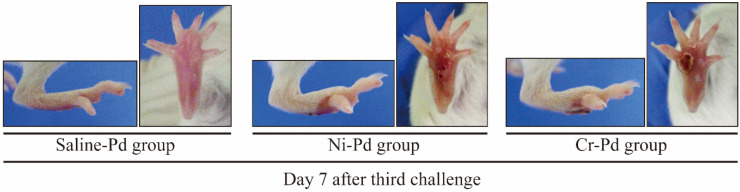
Macroscopic findings in the mouse model of cross-reactive metal allergy. At 7 days after the third challenge, the footpad swelling in the Cr-Pd and Ni-Pd groups was significantly increased compared with Saline-Pd group. Swelling and crust formation of the footpad were observed at 7 days after the third challenge in the Ni-Pd and Cr-Pd groups.

**Figure 3 ijms-21-04061-f003:**
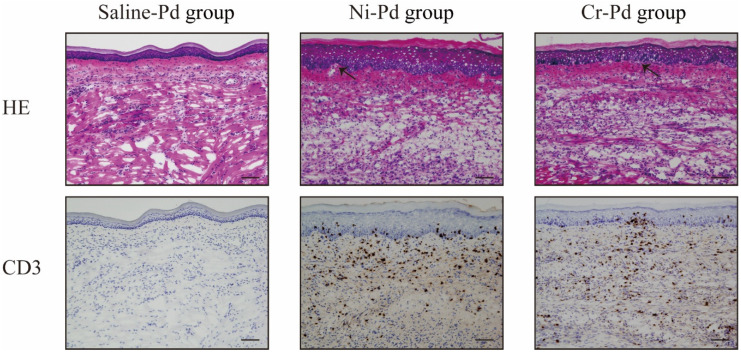
Histopathology and immunohistochemical analyses in the mouse model of cross-reactive metal allergy. At 7 days after the third challenge, Hematoxylin and Eosin (H&E) staining showed epithelial acanthosis, as well as epidermal spongiosis and liquefaction degeneration (arrows) in Ni-Pd and Cr-Pd groups. CD3^+^ T cells were also present in the epithelial basal layer and the upper dermis in the Ni-Pd and Cr-Pd groups. Scale bar = 10 µm.

**Figure 4 ijms-21-04061-f004:**
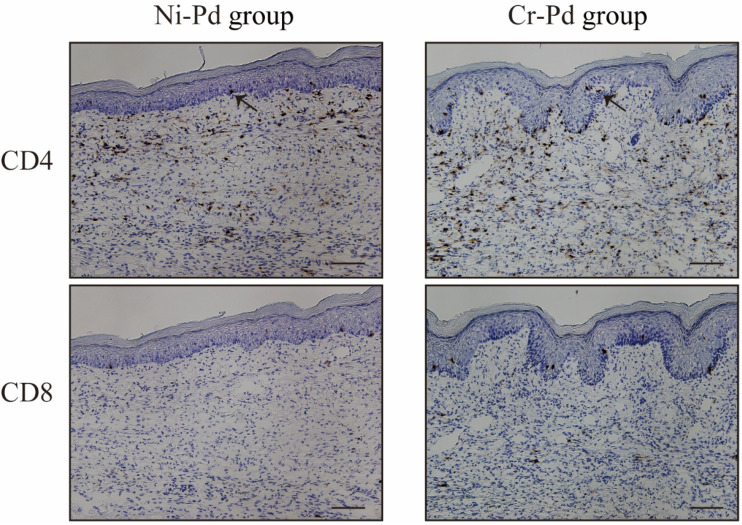
Immunohistochemical (IHC) analyses of CD4 and CD8 in the mouse model of cross-reactive metal allergy. CD3^+^ T cells infiltrated into the inflamed skin in the Ni-Pd and Cr-Pd groups. CD4^+^CD8^−^ T cells were present in the epithelial basal layer and the upper dermis (arrows). Scale bar = 10 µm.

**Figure 5 ijms-21-04061-f005:**
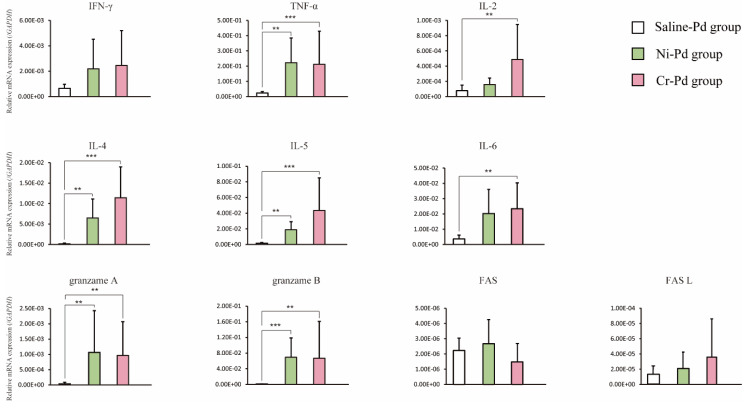
Expression levels of T cell cytokines, cytotoxic granules, and apoptosis-related genes in the mouse model of cross-reactive metal allergy. The expression levels of Th1 cytokines (interferon gamma [IFN-γ], tumor necrosis factor alpha [TNF-α], and interleukin [IL]-2), Th2 cytokines (IL-4, IL-5, and IL-6), cytotoxic granules (granzyme A and B), and apoptosis-related genes (Fas and Fas ligand [L]) were measured by qPCR analysis in the Ni-Pd Cr-Pd and Saline-Pd groups. Bars and error bars indicate the mean plus standard deviation. Statistical significance was tested by Dunn’s test (* *p* < 0.05, ** *p* < 0.01, *** *p* < 0.001).

**Figure 6 ijms-21-04061-f006:**
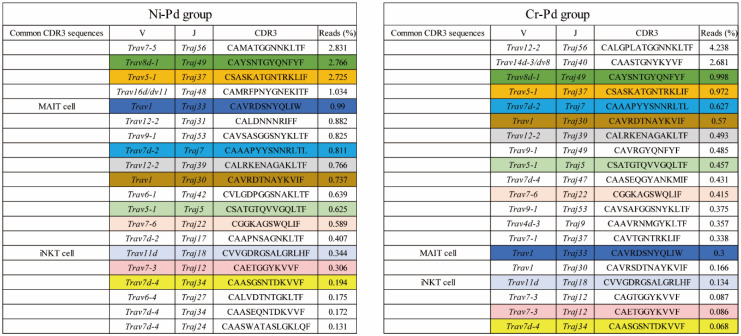
Distributions of the read (%) of amino acid sequences of CDR3 regions of TRAV and TRAJ in the mouse model of cross-reactive metal allergy. Amino acid sequences of CDR3 regions in the Ni-Pd and Cr-Pd groups at 7 days after the third challenge. T cells bearing *Trav8d-1-Traj49* and *Trav5-1-Traj37* occupied a high proportion of cells in the Ni-Pd and Cr-Pd groups. Shared T cells bearing the same TRAV, TRAJ, and CDR3 sequences colored in Figure 6 were obtained from both groups. Mucosal-associated invariant T (MAIT) cells and invariant natural killer T (iNKT) cells were also observed in both groups.

**Figure 7 ijms-21-04061-f007:**
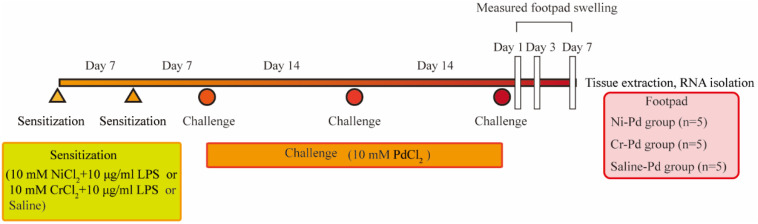
Schedule of the sensitization and challenge of the metal allergy cross-reactive mouse model. Sensitization using NiCl_2_ + lipopolysaccharide (LPS) or CrCl_2_ + LPS or saline were injected twice (at an interval of 7 days) intradermally (i.d.) into the left and right groin of mice. At 7 days after sensitization, mice were challenged for the first time. PdCl_2_ was injected into the left and right footpad by i.d. injection to challenge and elicit an immune reaction. Challenge was repeated three times at an interval of 14 days.

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
