# Peer review of "Cross-Reactivity of Palladium in a Murine Model of Metal-induced Allergic Contact Dermatitis"

_ijms, 2020, doi:10.3390/ijms21114061_

Round 1

Reviewer 1 Report

The manuscript ijms-814179 has been focused on cross-reactivity of palladium investigated in a murine model of metal-induced allergic contact dermatitis. Metal allergy is an important health problem.

Point by point remarks to the Authors

- In the title of the manuscript, as well as in the Abstract section the only metal mentioned is palladium. It seems necessary, at least in the Abstract, to provide the names of other metals. Moreover, the names of all 3 metals might be included in the keywords.

- The aim of the study has not been clearly described. In lines 67-69 it can be read what was done; however, the aim has not been clearly specified.

- Figure 1 presenting the schedule of sensitization and challenge of the metal cross-reactive mouse model should be presented in the Materials and Methods section. Typing errors are present in this figure. There is “Callenge”, but it should be “Challenge”. Moreover, in some places spaces are lacking (10 mM, instead of 10mM and the like).

- Figures 4 and 5: the Reviewer would like to suggest the Authors to indicate using arrows (for example) the changes presented in these figures. It would be very useful for a reader.

- In the Discussion section it seem to be necessary to put more attention to the practical/ clinical implications of this study.

- Line 234; Although chemicals names of PdCl2, CdCl2 and NiCl2 are commonly known in a scientific environment, it seems necessary to explain them.

Line 282; H2O2 (not H2O2).

Author Response

To reviewer 1

In the title of the manuscript, as well as in the Abstract section the only metal mentioned is palladium. It seems necessary, at least in the Abstract, to provide the names of other metals. Moreover, the names of all 3 metals might be included in the keywords.

Response:

As pointed out by the reviewer, we have changed abstract section (lines 29-34) and 3 metals included in the keywords (line 42).

The aim of the study has not been clearly described. In lines 67-69 it can be read what was done; however, the aim has not been clearly specified.

Response:

The aim of this study was to investigate the immune responses of T cells during the elicitation phase of allergic inflammation related to cross-reactivity between different metals. We have added the description of this in the Introduction section as follows (lines 71-74): “In the current study, we generated a novel mouse model of cross-reactive metal allergy. In addition, we investigated the immune responses (phenotypic markers, TCR repertoires, and cytokine expressions) of footpad-infiltrating T cells during the elicitation phase of allergic inflammation related to cross-reactivity between different metals.”

Figure 1 presenting the schedule of sensitization and challenge of the metal cross-reactive mouse model should be presented in the Materials and Methods section. Typing errors are present in this figure. There is “Callenge”, but it should be “Challenge”. Moreover, in some places spaces are lacking (10 mM, instead of 10mM and the like).

Response:

As suggested by the reviewer, we have changed Figure 1 to Materials and Methods section (line 259). In addition, we corrected the points that were pointed out (line 273).

Figures 4 and 5: the Reviewer would like to suggest the Authors to indicate using arrows (for example) the changes presented in these figures. It would be very useful for a reader.

Response:

As pointed out by the reviewer, we have changed Figure 4 (line 114) and 5(line 120).

In the Discussion section it seem to be necessary to put more attention to the practical/ clinical implications of this study.

Response:

Our results can provide a platform for the development of new diagnostic and treatment approaches relevant to the resolution of metal-induced allergies. We have added a description of the clinical implications in the Discussion section (lines 213-217).

Line 234; Although chemicals names of PdCl2, CrCl2 and NiCl2 are commonly known in a scientific environment, it seems necessary to explain them.

Response:

As suggested by the reviewer, we have added the details of chemicals names of PdCl2, CrCl2 and NiCl2 (line 241).

Line 282; H2O2 (not H2O2).

Response:

As suggested by the reviewer, we corrected the points that were pointed out (line 297).

Reviewer 2 Report

The study "Cross-reactivity of palladium in a murine model of metal-induced allergic contact dermatitis" conducted by Shigematsu, et al describes an interesting mice model of metal cross reactive allergic contact dermatitis. The immune responses by IHC analysis, cytokine measurements and TCR repertoire analysis are well done to explain the pad inflammation. I recommend its publication, but I also wish to point out the need for a minor correction: 

1)In Figure 1, challenge was misspelled several times (4) in the schedule of sensitization chart.  

2) Can the authors give an explanation why the Th1- and Th2-types of cytokines, as well as the cytotoxic granules are all affected equally in the model? Is that caused by LPS in the sensitization stage?

Although my last question should not require any editing of the manuscript, I am nevertheless curious about the following point: when the TCR repertoire is studied in two metal-treated groups, I wonder how easy it will be to correlate the positive findings on TRAV and TRAJ in mice to human patients with Pd or Ni contact dermatitis. I believe the authors might wish simply to mention the sharing mechanism at play in the two allergic responses.

Author Response

To reviewer 2

In Figure 1, challenge was misspelled several times (4) in the schedule of sensitization chart.

Response:

As suggested by the reviewer, we have corrected the points that were pointed out (line 273).

Can the authors give an explanation why the Th1- and Th2-types of cytokines, as well as the cytotoxic granules are all affected equally in the model? Is that caused by LPS in the sensitization stage?

Response:

Kitagaki et al. reported the repeated elicitation of contact hypersensitivity induced a shift in cutaneous cytokine milieu from a T helper cell type 1 to a T helper cell type 2 profile. [H Kitagaki et al; J Immunol 1997]. Our results were consistent with previous study thus we have discussed regarding these points(lines 178-181).